# Damage-Associated Molecular Patterns Modulation by microRNA: Relevance on Immunogenic Cell Death and Cancer Treatment Outcome

**DOI:** 10.3390/cancers13112566

**Published:** 2021-05-24

**Authors:** María Julia Lamberti, Annunziata Nigro, Vincenzo Casolaro, Natalia Belén Rumie Vittar, Jessica Dal Col

**Affiliations:** 1INBIAS, CONICET-UNRC, Río Cuarto, Córdoba 5800, Argentina; nrumievittar@exa.unrc.edu.ar; 2Department of Medicine, Surgery and Dentistry ‘Scuola Medica Salernitana’, University of Salerno, Baronissi, 84081 Salerno, Italy; annnigro@unisa.it (A.N.); vcasolaro@unisa.it (V.C.)

**Keywords:** miRNA, immunogenic cell death, cancer

## Abstract

**Simple Summary:**

Inside the cell, damage-associated molecular pattern molecules (DAMPs) play several physiological functions, but when they are released or translocated to the extracellular space, they gain additional immunogenic roles. Thus, DAMPs are considered key hallmarks of immunogenic cell death (ICD) in cancer, a functionally unique regulated form of stress-mediated cell death that activates the immune system response against tumor cells. Several epigenetic modulators of DAMPs have been reported. In this review, we aimed to provide an overview of the effects of microRNAs (miRNAs) on the expression of DAMPs and the putative link between miRNA, DAMPs, and cell death, focused on ICD. Overall, we propose that miRNAs, by targeting DAMPs, play critical roles in the regulation of both cell death and immune-associated mechanisms in cancer, while evidence of their potential involvement in ICD is limited. Finally, we discuss emerging data regarding the impact of miRNAs’ modulation on cancer treatment outcome.

**Abstract:**

Immunogenic cell death (ICD) in cancer is a functionally unique regulated form of stress-mediated cell death that activates both the innate and adaptive immune response against tumor cells. ICD makes dying cancer cells immunogenic by improving both antigenicity and adjuvanticity. The latter relies on the spatiotemporally coordinated release or exposure of danger signals (DAMPs) that drive robust antigen-presenting cell activation. The expression of DAMPs is often constitutive in tumor cells, but it is the initiating stressor, called ICD-inducer, which finally triggers the intracellular response that determines the kinetics and intensity of their release. However, the contribution of cell-autonomous features, such as the epigenetic background, to the development of ICD has not been addressed in sufficient depth. In this context, it has been revealed that several microRNAs (miRNAs), besides acting as tumor promoters or suppressors, can control the ICD-associated exposure of some DAMPs and their basal expression in cancer. Here, we provide a general overview of the dysregulation of cancer-associated miRNAs whose targets are DAMPs, through which new molecular mediators that underlie the immunogenicity of ICD were identified. The current status of miRNA-targeted therapeutics combined with ICD inducers is discussed. A solid comprehension of these processes will provide a framework to evaluate miRNA targets for cancer immunotherapy.

## 1. Introduction

Under physiological homeostasis, cell death involved in the continuous cellular turnover is non-immunogenic or even tolerogenic. This silence mechanism is imperative given that the activation of an immune response against dead cell-associated antigens would have autoimmune-related catastrophic consequences [1]. On the other hand, the death of infected or tumor cells under specific treatment conditions can elicit a robust antigen-specific immune response. This type of death that enhances the immunogenic potential of dying cells has been termed since 2005 as Immunogenic Cell Death (ICD) [2,3]. In the context of cancer therapies, several chemotherapeutics (doxorubicin, cisplatin, oxaliplatin), radiotherapy, oncolytic virus therapy, and photodynamic therapy have been characterized as ICD-inducers [4]. Innate and adaptive immune responses elicited by such anti-cancer agents are now deemed essential for an optimal therapeutic outcome, highlighting the clinical relevance of ICD.

Formerly, ICD was exclusively described in terms of immunogenic extrinsic or intrinsic apoptosis, but recently, with the increasing knowledge of cell death mechanisms, many non-apoptotic cell death processes have been involved in immune activation, including necroptosis, pyroptosis, and ferroptosis [5]. However, ICD cannot be considered only as a cellular death event given its strong dependence on the complex cellular communication between dying and immune cells [6]. In fact, for ICD to be successfully promoted, an exact combination of damage-associated molecular patterns (DAMPs) must be released/exposed in a spatiotemporally coordinated sequence and recognized by immune cells [7]. DAMPs are molecules that exert intracellular physiological functions, but gain additional immunogenic properties when they are exposed to the extracellular environment. Thanks to this peculiarity, DAMPs have been defined as ICD hallmarks. In this context, the immunogenicity of dying tumor cells is dependent on two main factors: antigenicity (conferred by tumor-associated antigens, cancer-testis antigen, and/or neo-epitopes) and adjuvanticity (provided by DAMPs) [8,9]. Accordingly, the inhibition of various processes involved in DAMPs’ expression, emission, and sensing could support the neoplastic cell full escape from immune recognition and elimination and potentiate malignant lethal progression.

Interestingly, several epigenetic modulators that suppress or activate genes encoding ICD-associated DAMPs have been discovered. They cause variations in gene expression through chromatin remodeling mechanisms, for example, DNA methylation and histone modification, and the generation of non-coding RNAs (ncRNAs), including long non-coding RNAs (lncRNAs), circular RNAs (circRNAs), and microRNAs (miRNAs) [10].

A recent review from Cruickshank et al. [8] recapitulated the evidence linking epigenetic modifications and ICD. In this review, our efforts are focused on extending that previous work by providing an overview of the current knowledge of miRNAs targeting ICD hallmarks and, thus, their potential involvement in ICD regulation and outcomes. We present a summary of the known miRNAs involved in DAMP modulation within tumors. Finally, we contextualize these findings in the design of novel therapeutic strategies.

Accumulating evidence indicates that several miRNAs can act as oncogenes (here termed oncomiRNAs) or tumor suppressor miRNAs by directly targeting different DAMPs, whose expression is frequently altered across human cancers. These observations lead us to forming the initial hypothesis that reestablishing the normal regulation of DAMP expression would constitute an important and clinically relevant goal to restore immunogenicity and to counteract immune evasion, which often results in the acquisition of chemoresistance and in tumor progression. However, as often happens in biological processes, the same molecules can perform completely different functions depending on the spatiotemporal context and/or the stimuli/insults to which the cell is subjected. Hence, DAMPs can both suppress cell death and promote the immunogenicity of cancer cells, according to tumor microenvironmental conditions. Therefore, we aimed here to provide a comprehensive and exhaustive overview of the effects of miRNAs on the expression of DAMPs, to help identify commonalities and differences that indicate possible interrelationships between miRNA, DAMPs, and cell death, with a special focus on ICD.

## 2. Immunogenic Cell Death

According to current models, cells undergoing ICD release or expose DAMPs on their surface can function as adjuvants for the innate immune system activation [9]. Most of these molecules have primarily non-immunological roles in the intracellular compartment before their mobilization by ICD. The adequate immunogenic response relies on the ability of specific stimuli to damage cells lethally while inducing the spatiotemporally coordinated emission of those DAMPs [11].

At the very beginning of the well-described ICD sequence, in a pre-apoptotic stage, the endoplasmic reticulum (ER) chaperone calreticulin (CRT) is translocated to the membrane before the cells exhibit phosphatidylserine residues [12]. Ecto-CRT mobilization works as an immunogenic “eat-me” signal for dendritic cells (DCs). This step is essential for an appropriate induction of ICD [12,13]. ICD also involves autophagy for optimal ATP active secretion. This autophagy-dependent release of ATP requires accumulation of ATP in autolysosomes [14]. In the extracellular space, ATP acts as a “find-me” signal for DCs and the consequent activation of the inflammasome to promote the release of the pro-inflammatory cytokine, interleukin (IL)-1β [15,16]. Doxorubicin [17] and PDT-treated cancer cells [18] also upregulate a type I interferon (IFN-1) signaling cascade resulting in C-X-C motif chemokine ligand 10 (CXCL10) secretion and finally in DC maturation. Although they are usually located inside the cells, heat shock proteins 70 and 90 (Hsp70 and Hsp90), under ICD, can be expressed at the surface of the cell membrane or released in the extracellular microenvironment and participate in immune stimulation [11,19]. In the late stages of apoptosis, high-mobility group box 1 (HMGB1) is passively released and binds Toll-Like Receptor 4 (TLR4) on DCs to increase antigen presentation [20].

The main objective of inducing ICD is to overcome the immunosuppressive phenotype of the tumor microenvironment through the restitution of the three signals between DC-T cell interaction, all of them mandatory for immunogenic T cell activation: (a) signal 1: antigen presentation; (b) signal 2: co-stimulation; and (c) signal 3: production of stimulatory cytokines [21]. Along this process, DCs engulf fragments of the stressed/dying cell and incorporate antigenic peptides into MHCs (antigenicity). During antigen presentation, the maturation signals triggered by DAMPs (adjuvanticity) lead to optimal activation of T cells, which finally detect and eliminate cancer cells in a highly precise, antigen-specific fashion [22] (Figure 1). 

Linking the immunogenic potential of ICD with an immunotherapy regimen is a promising approach for antitumor treatment. In this sense, we have recently summarized the findings suggesting the use of ICD as a strategy to optimize the current vaccine design for cancer immunotherapy [4,23]. Unfortunately, immunosuppression exerted by the tumor microenvironment limits the potential success of this strategy. In line with these observations, we propose here to identify which immune-activating or immunosuppressive ICD hallmarks are epigenetically targeted by miRNAs.

## 3. Epigenetic Regulation by miRNA

Over the last years, many studies have been conducted that support the idea that genetic information can be tightly regulated by non-coding RNAs (ncRNAs). ncRNAs do not code for proteins, and can be classified into long non-coding RNAs (lncRNAs), circular RNAs (circRNAs), and microRNAs (miRNAs). Among these ncRNAs, here we focus on post-transcriptional regulation of the expression of ICD molecular mediators by miRNAs.

The main function of miRNAs is to repress protein production, working as post-transcriptional regulators of mRNA. The miRNA biogenesis initiates with the generation of a large primary transcript (pri-miRNA), mainly transcribed by RNA polymerase II, which is 5′ capped and 3′ polyadenylated. The pri-miRNAs are then cleaved into precursor miRNAs (pre-miRNAs) that consist of around 85 nucleotides exhibiting a stem-loop structure. This cleavage is made by a microprocessor complex, composed of the RNA-binding protein DGCR8 and the type III RNase Drosha. Pre-miRNAs are then transported from the nucleus to cytoplasm by the Ran/GTP/Exportin 5 complex, where they are processed by another RNase III enzyme, Dicer, to a 20–22-nucleotide miRNA:miRNA* duplex. The * denotes the passenger strand, which is degraded, while the other complementary strand is the mature or guide strand. The mature/guided miRNA is then incorporated into a protein complex termed RNA-induced silencing complex (RISC) and guides RISC to target mRNA. miRNAs exert their effects by complementary base-pair binding to a short 7–8 nucleotide “seed” region typically located in the 3′ untranslated region (UTR) of the mRNA that they inhibit [24]. Importantly, one miRNA may regulate many targeted genes, while one gene may be targeted by many miRNAs [25].

Nowadays, it is well known that transcriptional control changes, chromosomal abnormalities, epigenetic changes, and defects in the miRNA biogenesis machinery may lead to an aberrant miRNA expression in human cancers. It has been described that this dysregulation affects one or more of the hallmarks of cancer described by Hanahan and Weinberg [26]. Thus, in a cancer context, depending on their target genes and the environmental conditions, miRNAs could function as either oncogenes (termed here oncomiRNAs) or tumor suppressors [27]. 

Current evidence sheds more light on the functional properties of miRNAs and opens new paradigms that need to be further explored [25]. In this sense, it has been demonstrated that miRNAs can act in different cellular locations (cytoplasm, mitochondrion, nucleus, and exosomes) [28,29,30] and even bind their targets in different binding sites (5′UTR, coding region, and 3′UTR) [31,32]. Several additional non-canonical binding clusters independent of seed region have been discovered [33]. It has also been reported that, under certain circumstances, instead of repression, miRNAs activate their target expression [34]. Finally, possible interactions have been observed between other ncRNAs and miRNA:mRNA complexes [35]. 

## 4. Epigenetic Regulation of ICD-Hallmarks by miRNA

Given their well-known contributions to cell death control [36] and autophagy [37], miRNAs may easily be envisioned to play key roles in the processes regulating ICD. 

In the following section, we recapitulate the current data supporting epigenetic regulation of DAMPs by miRNAs. In particular, we have only focused on studies in which direct miRNA:mRNA interactions have been experimentally validated, e.g., by gene reporter or target protector-mediated assays [38]. Given that both miRNAs and DAMPs roles are so much context-dependent, we propose a classification of miRNA function (as tumor suppressor or oncomiRNA) solely based on factual experimental evidence. Surprisingly, we notice that they mainly function as tumor suppressors. For miRNA nomenclature, we decided to adapt and unify miRNA names and accession numbers according to the miRBase database, the primary repository for published miRNA, freely available at http://www.mirbase.org/ (accessed on 25 January 2021). All data are summarized in Table 1 and schematized in Figure 2.

### 4.1. Calreticulin

Calreticulin (CRT) is a chaperone protein mainly located in the lumen of the endoplasmic reticulum (ER). Given its high capacity of buffering calcium, it modulates calcium signaling and homeostasis [83]. Through these functions, CRT has important biological regulatory roles inside and outside the ER, and is involved in cancer, wound healing, cardiogenesis, autoimmune diseases, and neurological diseases [84]. 

In addition, CRT can be translocated to the surface of stressed and dying cells [12]. Exposure of CRT on the cell surface is a major factor in ICD, as it serves as an “eat me” engulfment signal for DCs, thus promoting the presentation of tumor-associated antigens to T cells [12]. In the context of cancer, CRT was identified as a direct target of miR-27a-3p. The role of this miRNA seems controversial, as it has been previously assigned with both anti-tumor [85,86,87] and pro-tumor properties [88,89]. miR-27a-3p downregulates CRT expression by inhibiting target mRNA translation [67]. It was also demonstrated that the miR-27a-3p throught targeting CRT modulates MHC class I surface exposure, and that, in particular, high miR-27a-3p concomitant with low CRT expression associates with enhanced tumor growth in vivo, colorectal cancer stage, development of metastasis, and impairment of CD8^+^ T-cell infiltration [67]. When colorectal cancer cells were subjected to ICD inducers (mitoxantrone and oxaliplatin), miR-27a-3p blocked CRT exposure, as well as ATP and HMGB1 secretion. Upon chemotherapeutic treatment, miR-27a-3p levels were inversely correlated with induction of apoptosis (by ICD inducers) and autophagy (by chloroquine). In parallel, soluble factors released by miR-27a-3p overexpressing tumor cells subjected to ICD failed to induce DC functional and phenotypic maturation [68]. The investigations of Colangelo et al. [67,68] discussed in this section support the idea that the miR-27a-3p/CRT axis modulates the ICD program, especially by blocking the initial interaction between DCs and ICD-subjected tumor cells.

Collectively, data recapitulated here suggest that miR-27a can be postulated as an oncomiRNA in colorectal cancer, where its expression was shown to be upregulated, and that it may support tumor progression through downregulation of CRT-dependent immunostimulation.

### 4.2. Heat Shock Proteins

Heat-shock proteins (Hsps) are a group of molecular chaperones whose main cellular function is to ensure the precise (re)folding of proteins in stress conditions. They are usually located in the intracellular space wherein they exert prominent cytoprotective functions. Importantly, many tumors overexpress Hsps, presumably as an adaptive response to a “stressful” niche where they develop [90]. However, the ability of Hsps to contribute chaperoned peptides for antigen processing and MHC-restricted presentation has not yet been elucidated [91].

Under some circumstances, for example during ICD, at least two members of this family, Hsp70 and Hsp90, can be expressed at the cell surface, where they exhibit immunostimulatory properties [92,93]. Therefore, miRNAs that target Hsps expression are likely to have a significant impact on the tumor phenotype. 

In pancreatic ductal adenocarcinoma, miR-142-3p inhibited cell proliferation by negatively regulating Hsp70 expression [53]. In addition to its proliferative role in cancer, Hsp70 was associated with chemoresistance. In this context, it was reported that miR-223-5p suppressed the chemoresistance of osteosarcoma cells to cisplatin through JNK/Jun signaling by downregulating Hsp70 expression; however, expression of miR-223-5p was reduced in osteosarcoma biopsies compared with paired non-tumor tissues [64]. Interestingly, also in an in vitro osteosarcoma model, ectopic expression of miR-223-3p, through Hsp90 downregulation, inhibited cell proliferation by inducing cell cycle arrest and apoptosis [63]. miR-27b-3p and miR-628-3p directly targeted Hsp90, resulting in suppression of non-small-cell lung carcinoma migration and invasion, and promoted apoptosis [69,81]. In addition, the lncRNA KCNQ1OT1 modulated Hsp90 expression by blocking miR-27b-3p [69]. The expression of miR-361-5p in cervical cancer was also downregulated by a lncRNA NEAT1, whereas its negative modulation of Hsp90 inhibited invasion and epithelial-mesenchymal transition (EMT) [73].

miRNAs described here act on the overall expression of Hsps, but the effects on postransductional modification and surface translocation are not revealed. However, the potential contribution of these miRNAs in modulating Hsps expression in the context of ICD remains an unmet question; thus, it is not yet elucidated whether they can play in favor of or against immunogenicity.

### 4.3. HMGB1

HMGB1 is a non-histone chromatin-binding protein that regulates different cellular functions according to its cellular localization. Within the nucleus, HMGB1 is involved in many DNA events (repair, transcription, stability, telomerase maintenance). In the cytoplasm, membrane, or extracellular space, several studies have demonstrated its ability to regulate cell proliferation, apoptosis, autophagy, inflammation, invasion, metastasis, and immunity, among others [94]. Paradoxically, HMGB1 has been attributed with both pro- and anti-tumoral properties. Relative to its immune functions, HMGB1 has been described as both a suppressor and an activator, which depends on receptors, targeted cells, and redox state [95]. 

A series of studies have demonstrated that miRNAs participate in the regulation of HMGB1 expression. By targeting HMGB1, miR-548b-3p [80] and miR-320-3p [65] suppressed hepatocellular carcinoma (HCC) cell proliferation, metastasis, and invasion while inducing apoptosis. In addition, HMGB1 expression was downregulated in HCC specimens and cell lines, which correlated with poor prognosis. Along this line, down-regulation of miR-325 [66] and miR-449a [74] correlated with poor prognosis in lung cancer patients, as these miRNAs negatively targeted HMGB1, resulting in decreased cell migration, invasion, and/or proliferation. It was also discovered that miR-107 [40] and miR-1179 [41], whose expression was downregulated in breast cancer and gastric cancer, respectively, inhibited autophagy, proliferation, and/or migration of tumor cells by directly suppressing HMGB1. While miR-1284 downregulation in cervical cancer tissues and cell lines correlated with poor survival, the miR-1284/HMGB1 axis suppressed proliferation and invasion [42]. Restoration of miR-665, by directly targeting HMGB1, suppressed cell proliferation, colony formation, migration, and invasion, and induced cell apoptosis in retinoblastoma [82]. Similar effects of let-7e-5p were observed in thyroid cancer, whereas similar effects were observed in thyroid cancer but through let-7e-5p [39].

Both mature sequences of mir-34a were shown to target HMGB1 directly. Antitumor events were promoted by the miR-34a-5p/HMGB1 axis in acute myeloid leukemia [96], cutaneous squamous cell carcinoma [72], cervical cancer, and colorectal cancers [71], and similar effects were exerted by miR-34a-3p/HMGB1 in retinoblastoma [70]. In acute myeloid leukemia, miR-181b-5p was significantly decreased, especially in relapsed/refractory patients. Upregulation of miR-181b-5p increased the chemosensitivity of leukemia cells and promoted drug-induced apoptosis via negative modulation of HMGB1 expression [54]. When a miR-200c-3p mimic was transfected into lung [58] and breast cancer cells [57], there was a significant decrease in cell migration, invasion, and epithelial to mesenchymal transition (EMT). These changes were associated in part with the downregulation of HMGB1 by miR-200c-3p. In triple negative breast cancer, the downregulation of miR-205-5p was negatively associated with progression and metastasis, and cell growth and EMT were inhibited by the miR-205-5p/HMGB1 axis [59].

The antitumor role of the miR-129-5p/HMGB1 axis was studied in osteosarcoma [45], hepatocellular carcinoma [48], breast cancer [43,44], gastric cancer [46,47], and colon cancer [45]. Downregulation of this miRNA expression was also reported in cancer cells lines and primary cancers, compared to their normal counterparts [46]. In these reports, ectopic expression of miR-129-5p was found to suppress migration, invasion, proliferation, and EMT, while it enhanced apoptosis, radio-, and chemosensitivity through HMGB1 downregulation. In addition, the miR-505-3p/HMGB1 axis exerted a negative impact on tumor progression in osteosarcoma [78], hepatocellular carcinoma [76,77], and gastric cancer [75].

Interestingly, it was found that the miRNA/HMGB1 axis was modulated by long noncoding RNAs (lncRNAs) and circular RNAs (circRNAs), most of them upregulated in cancer tissues. The lncRNA prostate cancer-associated transcript 1 (PCAT-1) [48] and MALAT1 [45,49] inhibited reversed miRNA-dependent HMGB1 downregulation. Likewise, the lncRNA UCA1 exerted pro-tumoral activity in lung cancer, acting mechanistically by upregulating HMGB1 expression through miR-193a-3p inhibition [55]. The lncRNA PCA3 [62] downregulated the expression of miR-218-5p, whose negative targeting of HMGB1 was evaluated in endometrial carcinoma [60], lung cancer [61], and prostate cancer [62], leading to a decrease in tumor proliferation, migration, invasion, and chemoresistance. miR-142-3p has also been identified as a microRNA that suppressed HMGB1 expression in non-small-cell lung carcinoma [51,52] and glioma [50], therefore playing a major role in reducing tumor cell proliferation, invasion, apoptosis, and drug resistance (to cisplatin). Along this line, the lncRNA TP73-AS1 upregulated HMGB1 expression through sponging miR-142-3p [50] and also miR-200a-5p [56]. Finally, the circRNA circ_0007385 enhanced cell proliferation, migration, invasion, and chemoresistance in lung cancer through upregulating miR-519d-3p and thus downregulating HMGB1 [79].

Overall, the data recapitulated in the last two sections suggest that, at least in the described phenotypes, the cytoprotective roles of HMGB1, Hsp70, and Hsp90 were more relevant than their immunostimulatory properties. Indeed, their targeting miRNAs appeared to repress pro-tumoral phenotypes by directly inhibiting their expression, thereby functioning as potential tumor suppressors. Moreover, some of these miRNAs were shown to be downregulated in tumors compared to their normal counterparts (Table 1). These findings might enhance our knowledge of the molecular mechanisms underlying malignant progression, making miRNAs promising targets for therapeutic intervention.

## 5. Modulation of Therapeutic Outcome through Targeting DAMPs by miRNA

Nowadays, drug resistance represents a main obstacle in the clinical setting, leading to relapse and metastasis in several cancer types. Therefore, new and more innovative approaches are required to treat these malignancies efficiently. The new knowledge on miRNA molecular actions and their involvement in cancer-associated mechanisms has opened new perspectives in the development of more effective therapeutic strategies. As miRNAs modulate multiple signaling pathways associated with therapy response, modification in miRNA expression can lead to significant changes in disease evolution and cancer outcome. Over the last few decades, numerous studies have been published on miRNA regulation of the cancer treatment response [97]. Compelling evidence has shown that the fundamental mechanisms of resistance to different anticancer drugs might be attributed to aberrantly expressed miRNAs in a wide range of malignancies [98]. Some miRNAs that have been described above as tumor suppressors have also been shown to modulate the sensitivity to chemotherapeutic agents.

By directly targeting HMGB1, several miRNAs may synergistically promote chemosensitivity, increasing drug pro-apoptotic activity often impaired by HMGB1-promoted autophagy. Of those, Lu et al. reported that tumor cells exhibit greatly enhanced apoptosis-related sensitivity to doxorubicin or cytarabine after transfection with the miR-181b mimic [54], and, in line with this, Chen & Li showed that miR-1284 enhanced cisplatin-induced apoptosis [42]. Ectopic expression of miR-505-3p was shown to enhance doxorubicin-induced cell death and caspase-3 dependent apoptosis, via inactivation of the Akt pathway [77]. The PI3K/Akt/mTOR pathway has been implicated in HMGB1-mediated autophagy, which was shown to play an important pro-survival role and contribute to chemoresistance [99]. Moreover, Chen et al. reported that miR-142-3p overexpression inhibited autophagy by activating the PI3K/Akt/mTOR pathway through HMGB1, and thus resulting in the increase of cisplatin and doxorubicin-chemosensitivity [51]. In addition, HMGB1-mediated autophagic resistance to paclitaxel was identified by Ran et al., whereas upregulation of miR-218-5p could restore chemosensitivity [60]. Consistently, miR-129-5p was found to increase apoptosis during paclitaxel treatment, and the improvement in sensitivity was associated with inhibition of autophagy [44]. Liu et al. demonstrated that miR-34a-3p-mediated inhibition of autophagy could sensitize cells to etoposide and carboplatin-induced apoptosis [70]. Similarly, Tang et al. precisely described a feedback loop formed by miR-223-5p, Hsp70, and the JNK/Jun signaling pathway associated with the modulation of cisplatin-resistance [64].

Interestingly, most of the miRNAs with tumor suppressor functions are found downregulated in various cancer types compared to normal tissues: miR-505-3p in hepatocellular carcinoma [76] and osteosarcoma [78], miR-1284 in cervical cancer [42], miR-129-5p in osteosarcoma [49], miR-142-3p in non-small-cell lung cancer [52] and in osteosarcoma [49], and miR-223-5p in osteosarcoma [64], among others. This aberrant expression is closely implicated in cancer treatment resistance. However, even when the aberrant expression of those miRNA induced a substantial upregulation of ICD-associated DAMPs, none of the above-mentioned reports have explored its association with the induction of ICD. Elucidation of those features would require more targeted assays, which would allow us to obtain a more comprehensive view about the epigenetic modulation occurring in cancer cells undergoing ICD.

## 6. Therapeutic Combination of ICD and miRNAs: A New Opportunity

Innovative therapeutic approaches including miRNA-based agents together with current standard treatment modalities could importantly benefit cancer patients. Targeting miRNAs by restoring their expression seems to be an attractive tool for emerging, more effective individualized therapies. However, there are still significant challenges to overcome to ensure the efficient delivery of miRNA to the tumor in vivo, for example, off-target effects, poor serum stability, and ineffective, poorly selective cellular entry. Hence, the development of novel drug delivery systems with the capacity to target-directed transport and protection of such cargos is mandatory [100].

In the face of these challenges, two different delivery systems have been designed in order to combine miRNA and ICD-inducer administration.

Phung et al. successfully fabricated nanoparticles (NPs) for target-specific co-delivery of low-dose doxorubicin and miR-200c to cancer cells [101]. These mixed NPs were composed of two co-polymers: poly(d,l-lactide-co-glycolide)-polyethyleneimine (PLGA-PEI) and folic acid (FA)-conjugated PLGA-block-poly(ethylene glycol) (PLGA-PEG-FA). Doxorubicin was encapsulated by the hydrophobic core formed by PLGA, whereas the negative charges of miR-200c were electrostatically absorbed by the cationic backbone of PEI. PEI cytotoxicity was reduced by PEG, which also conferred stability and the way to introduce folic acid (FA) [102]. Despite conferring pH and temperature stability, permeability, non-immunogenicity, capacity to be tagged, among other advantages, FA is extremely important in this system because it may increase the proper and selective uptake by tumor cells in both in vitro and especially in vivo settings. This is because folate receptors are overexpressed in numerous cancer types compared to corresponding normal tissues [103]. This configuration was in fact successful because it protected miR-200c from degradation in serum. In addition, the complete NPs were well tolerated by hosts, probably due, at least in part, to the substantially low dose of doxorubicin used [101].

Surprisingly, although miR-200c has been shown to downregulate HMGB1 expression directly [57,58], this new NP-based platform that combined miR-200c and doxorubicin promoted ICD via the translocation of CRT to the cell surface and the release of HMGB1 in vitro and in vivo. Accordingly, DAMPs’ modulation was accomplished by DCs’ phenotypic maturation. In addition, T cell antitumor activity was enhanced, partially due to downregulation of PD-L1 by miR-200c [101].

Recently, Wang et al. designed another type of nanocarrier to deliver both miR-1284 and cisplatin [104]. They generated liposomes composed of distearoylphosphatidylcholine (DSPC) succinylphosphatidylethanolamine (DSPE-mPEG), distearoyl-N-(3-carboxy-propionoylpoly (ethyleneglycol) succinyl) phosphatidylethanolamine (DSPE-PEG-COOH), and 1,2-dioleoyl-3-trimethylammoniumpropane (DOTAP) to be loaded with 10% of cisplatin and surface-conjugated with the CD59 antibody. The presence of anti-CD59 on the NP surface conferred tumor selective accumulation properties, as CD59 has been shown to be overexpressed in cancer and associated with immune escape events [105]. This cationic liposome was able to bind the negatively charged miR-1284 electrostatically.

The pharmacokinetic analysis of the formulation indicated that the nanocarrier-based system prolonged the blood circulation of the drugs. The anticancer effects were only assessed in vitro and demonstrated a full internalization of the complex by tumor cells. As expected and according to previous results [42], co-delivery of miR-1284 and cisplatin synergistically inhibited cell viability and promoted apoptosis by HMGB1 downregulation [104]. Given these promising results, future in vivo analysis should be done to evaluate the pre-clinical efficacy of this formulation. The impact of this combination should also be studied in the context of ICD, given that cisplatin was shown to be an ICD inducer [4].

Together, these results provide guidance for a promising combination strategy to improve the clinical use and the immunostimulatory efficiency of ICD-inducing drugs and develop an effective and safe cancer therapeutic option.

## 7. Future Challenges

Understanding the regulation of ICD hallmarks is pivotal for a better exploitation of the different effects characterizing ICD inducers. In this review, we summarized studies that have reported that numerous miRNAs contributed to DAMP modulation, acting as either oncogenic miRNAs or tumor suppressor miRNAs. However, it is important to note that there is not much evidence regarding miRNA modulation by ICD, although it was reported that different miRNAs could contribute to ICD inducers therapeutic activities. Outstanding questions exist: Does a miRNA signature associated with ICD actually exist? Do specific miRNAs play a role in ICD in cancer cells?

As mentioned above, ICD-associated pathways are not tumor-exclusive. For that reason, when evaluating the role of an immunity modulator, it is necessary to integrate fully the different cellular actors and the molecular crosstalk established between them. In this scenario, it was demonstrated that miRNAs could be involved in the paracrine dialogue governing inter-cellular signaling within the tumor microenvironment. In this sense, dysregulation of endogenous miRNAs can be induced in immune cells by cancer cell-released DAMPs or miRNAs derived from cancer cells that can directly affect immune cell functions. Unlu et al. demonstrated that miR-34c and miR-214 were specifically upregulated in human PBMCs following incubation with conditioned media or tumor cell lysate from stressed cells, as part of the inflammatory response. In particular, the presence of HMGB1 within the paracrine stimulus was strongly associated with miRNA modulation [106]. Frank et al. provided evidence that miR-375 transfer from tumor cells to macrophages is crucial to alter the tumor-associated macrophage phenotype and the subsequent development of a tumor-promoting microenvironment [107]. Moreover, tumor-secreted miR-21 and miR-29a also can function by another mechanism, by binding as ligands to the Toll-like receptor (TLR) 8, in immune cells, triggering a TLR-mediated prometastatic inflammatory response that ultimately may lead to tumor growth and metastasis. Thus, by acting as paracrine agonists of TLRs, secreted miRNAs are key regulators of the tumor microenvironment and are implicated in tumor–immune system communication [108]. Intriguingly, recently, Lee J et al. generated in the laboratory two distinct modified single-stranded RNAs (ssRNAs) and showed their ability to induce immunogenic cell death in different types of cancer cell lines. In particular, those ssRNA promoted DAMP release and consequent impact on cytokine secretion by immature DCs [109]. We have recently discovered that the employment of immunogenic tumor cell lysates as a tumor antigen source in the development of DC-based vaccines influenced the miRNA profile in DCs. Therefore, we wonder: Could cancer cells undergoing ICD exploit specific miRNAs to communicate with the tumor immune microenvironment? Could cancer cell-derived miRNAs become messengers for DCs during ICD?

These questions compelled us to match several multifaceted interactions to understand the effects of environmental factors on miRNA expression more comprehensively. In this sense, a novel discipline termed “molecular pathological epidemiology” (MPE) has been proposed as a comprehensive approach to precision medicine. MPE protocols include a multilevel research platform to integrate molecular pathologies, immune response, and clinical outcomes in cancer. The application of in vivo pathology together with new multi-omics techniques might contribute to a more profound understanding of miRNA heterogeneous regulation, their role in tumor biology and therapeutic response, and the putative link with the endogenous/exogenous environment (diet, drugs, including ICD-inducing agents, microbiota, and germline genetics), further supporting the design of targeted, personalized therapies [110,111].

## 8. Conclusions

The discovery of miRNAs has deepened our knowledge of human diseases, especially cancer and its supporting epigenetic mechanisms. In this article, we have reviewed the evidence regarding the effects of miRNAs on the expression of DAMP mRNAs in different cancer types. Initially, we expected to identify several oncomiRNAs targeting DAMPs whose ectopic inhibition could promote and/or restore the immunogenic potential of therapeutic agents. Surprisingly, our hypothesis was only valid for CRT targeting, whereas miRNA-based downregulation of other DAMPs was in fact associated with suppressed cancer properties. We propose several reasons to explain, at least in part, these unexpected data. First of all, most DAMPs which were shown to be regulated by miRNAs are classified as constitutive (cDAMPs); hence, they are present inside healthy cells, and are released following immunogenic stimulus, exerting their proinflammatory functions in the extracellular space. Their ICD-associated modulation is mainly post-translational, whereas miRNA regulation is exerted at a pre-translational level. However, it seems possible that downregulation by miRNAs in cancer cells could temporarily limit their release and so their immunogenic roles. Further studies should be done in order to clarify this hypothesis. In contrast to cDAMPs, inducible DAMPs (iDAMPs) are not present in healthy cells but are induced and/or altered upon cell death (e.g., type 1 IFN). As far as we know, the regulation of iDAMPs by miRNAs has not yet been determined.

Secondly, when studying miRNAs, it is important to consider the relative promiscuity of their targets. As stated above, a given miRNA may have thousands of targets with significant differences in function. Even in the presence of restoration-of-function assays, this would lead to paradoxical conclusions about the role of miRNAs, in that a single miRNA may theoretically impact in opposite ways within the cell by targeting effectors with opposite functions. Moreover, interpretation of this paradox is even more complex when considering that miRNAs probably show different functions depending on the environment in which they are expressed.

However, given miRNAs’ role in regulating cellular processes as cell death and autophagy, but also different immune escape mechanisms modulating antigen processing/presentation and immune inhibitory molecules in different types of cancer cells, it remains reasonable to speculate their involvement in ICD. This is especially true if we consider that miRNAs are implicated in tumor–immune system communication, a key feature of ICD.

## Figures and Tables

**Figure 1 cancers-13-02566-f001:**
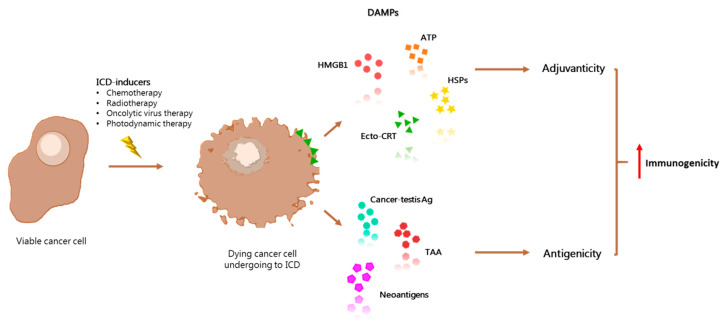
ICD-antitumor effect relies on the enhancement of adjuvanticity and antigenicity of tumor cells. In the tumor microenvironment, immunogenic cell death (ICD) triggered by several ICD-inducers plays a major role in stimulating antitumor immune response. Its lethal action leads to the release of tumor associated antigens (TAAs), cancer testis antigens, and neoantigens, which ultimately increases antigenicity. The concomitant exposure of damage-associated molecular patterns (DAMPs), such as ATP, calreticulin (CRT), HMGB1, and Hsps (Hsp70, Hsp90), confers a robust adjuvanticity to dying cancer cells. Both antigenicity and adjuvanticity enhancement leads to an exacerbated immunogenicity of ICD-undergoing tumor cells.

**Figure 2 cancers-13-02566-f002:**
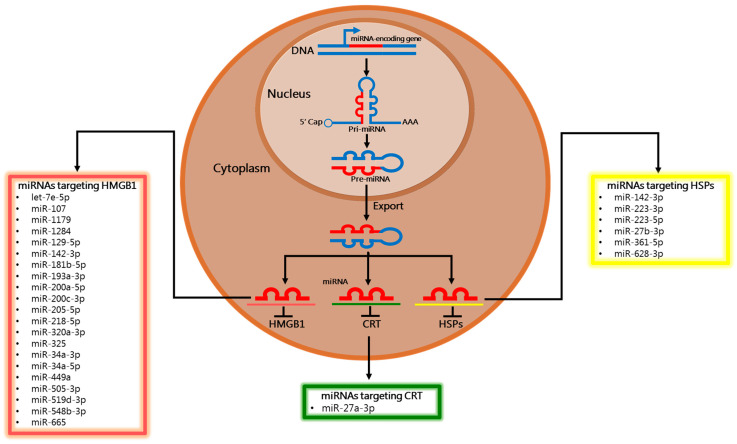
Immunogenic Cell Death-hallmarks targeted by miRNAs. The miRNA processing pathway initiates with the transcription of the primary miRNA (pri-miRNA) and its cleavage to generate the pre-miRNA. Then, pre-miRNA is exported from the nucleus to the cytoplasm, where it is cleaved to its mature miRNA. The functional strand of the mature miRNA has the major role to silence target mRNAs through different mechanisms. In this review, we summarize miRNAs that target well-known Immunogenic Cell Death-associated DAMPs, such as calreticulin (CRT), HMGB1, and Hsps (Hsp70, Hsp90).

**Table 1 cancers-13-02566-t001:** miRNA directly targets DAMPs and their tumor suppressor or oncomiRNA associated roles. According to the miRBase nomenclature, miRNAs are named with the abbreviation “miR” followed by a dash and a number. These are preceded by a prefix representing the species (for example, “hsa” for *Homo sapiens*). Each miRNA is also identified by its miRBase accession number and, if available, the type of its dysregulation reported in the tumor model analyzed, according to which it may be classified as tumor suppressor or oncomiRNA. For each targeted DAMP, the effects of its regulation are also described (↓ decrease/inhibition; ↑ increase/activation).

miRNA	miRBase Accession Number	Dysregulation in Cancer vs. Normal Counterparts	Target DAMP	miRNA Binding Site in Target Gene	Cancer Type	Effects	Role	Reference
hsa-let-7e-5p	MIMAT0000066	Not evaluated	HMGB1	3′UTR	Thyroid cancer	↓ Migration ↓ Invasion	Tumor suppressor	Ding et al., 2019 [39]
hsa-miR-107	MIMAT0000104	Downregulation (tissues and cell lines)	HMGB1	3′UTR	Breast cancer	↓ Migration ↓ Proliferation ↓ Autophagy	Tumor suppressor	Ai et al., 2018 [40]
hsa-miR-1179	MIMAT0005824	Downregulation (tissues and cell lines)	HMGB1	3′UTR	Gastric cancer	↓ Proliferation ↓ Invasion	Tumor suppressor	Li et al., 2019 [41]
hsa-miR-1284	MIMAT0005941	Downregulation (tissues and cell lines)	HMGB1	Not reported	Cervical cancer	↓ Proliferation ↓ Invasion ↑ Cisplatin-induced apoptosis	Tumor suppressor	Chen & Li, 2018 [42]
hsa-miR-129-5p	MIMAT0000242	Not evaluated	HMGB1	3′UTR	Breast cancer	↓ Irradiation-induced autophagy ↑ Radiosensitivity	Tumor suppressor	Luo et al., 2015 [43]
		Not evaluated		Not evaluated	Breast cancer	↓ Autophagy ↑ Paclitaxel-induced apoptosis	Tumor suppressor	Shi et al., 2019 [44]
		Not evaluated		Not reported	Colon cancer	↓ Proliferation	Tumor suppressor	Wu et al., 2018 [45]
		Downregulation (tissues and cell lines)		3′UTR	Gastric cancer	↓ Proliferation ↑ Apoptosis	Tumor suppressor	Feng et al., 2020 [46]
		Downregulation (cell lines)		3′UTR	Gastric cancer	↓ Proliferation ↓ Epithelial-mesenchymal transition	Tumor suppressor	Wang et al., 2019 [47]
		Not evaluated		3′UTR	Hepatocellular carcinoma	↓ Migration ↓ Invasion	Tumor suppressor	Zhang et al., 2017 [48]
		Downregulation (tissues)		3′UTR	Osteosarcoma	↓ Proliferation ↑ Apoptosis	Tumor suppressor	Liu et al., 2017 [49]
hsa-miR-142-3p	MIMAT0000434	Downregulation (tissues)	HMGB1	3′UTR	Glioma	↓ Proliferation ↓ Invasion	Tumor suppressor	Zhang et al., 2018 [50]
		Not evaluated		3′UTR	Non-small-cell lung cancer	↓ Starvation-induced autophagy ↑ Cisplatin and doxorubicin-chemosensitivity	Tumor suppressor	Chen et al., 2017 [51]
		Downregulation (tissues and cell lines)		3′UTR	Non-small-cell lung cancer	↓ Proliferation ↑ Apoptosis	Tumor suppressor	Xiao & Lu, 2015 [52]
		Downregulation (tissues)		3′UTR	Osteosarcoma	↓ Proliferation ↑ Apoptosis	Tumor suppressor	Liu et al., 2017 [49]
		Not evaluated	Hsp70	3′UTR	Pancreatic ductal adenocarcinoma	↓ Proliferation	Tumor suppressor	MacKenzie et al., 2013 [53]
hsa-miR-181b-5p	MIMAT0000257	Not evaluated	HMGB1	3′UTR	Acute myeloid leukemia	↑ Doxorubicin or cytarabine-induced apoptosis	Tumor suppressor	Lu et al., 2014 [54]
hsa-miR-193a-3p	MIMAT0000459	Not evaluated	HMGB1	3′UTR	Lung cancer	↓ Proliferation ↓ Migration	Tumor suppressor	Wu et al., 2018 [55]
hsa-miR-200a-5p	MIMAT0001620	Downregulation (tissues)	HMGB1	3′UTR	Hepatocellular carcinoma	Not determined	Tumor suppressor	Li et al., 2017 [56]
hsa-miR-200c-3p	MIMAT0000617	Not evaluated	HMGB1	3′UTR	Breast cancer	↓ Invasion ↓ Migration	Tumor suppressor	Chang et al., 2014 [57]
		Not evaluated		Not evaluated	Lung cancer	↓ Invasion ↓ Migration ↓ Epithelial-mesenchymal transition	Tumor suppressor	Liu et al., 2017 [58]
hsa-miR-205-5p	MIMAT0000266	Downregulation (tissues). Moreover, it is downregulated in metastatic compared to non-metastatic cancer	HMGB1	3′UTR	Triple-negative breast cancer	↓ Invasion ↓ Migration ↓ Proliferation	Tumor suppressor	Wang et al., 2019 [59]
hsa-miR-218-5p	MIMAT0000275	Downregulation Paclitaxel-resistant compared to non-drug resistant cells (cell lines)	HMGB1	3′UTR	Endometrial carcinoma	↓ Autophagy ↑ Paclitaxel-chemosensitivity	Tumor suppressor	Ran et al., 2015 [60]
		Not evaluated		3′UTR	Lung cancer	↓ Invasion ↓ Migration	Tumor suppressor	Zhang et al., 2013 [61]
		Non evaluated		3′UTR	Prostate cancer	↓ Invasion ↓ Migration ↓ Proliferation ↑ Apoptosis	Tumor suppressor	Zhang et al., 2019 [62]
hsa-miR-223-3p	MIMAT0000280	Not evaluated	Hsp90	3′UTR	Osteosarcoma	↓ Proliferation ↑ Apoptosis ↑ Cell cycle G0/G1 arrest	Tumor suppressor	Li et al., 2012 [63]
hsa-miR-223-5p	MIMAT0004570	Downregulation (tissues and cell lines)	Hsp70	3′UTR	Osteosarcoma	↑ Cisplastin-induced apoptosis	Tumor suppressor	Tang et al., 2018 [64]
hsa-miR-320a-3p	MIMAT0000510	Downregulation (tissues)	HMGB1	3′UTR	Hepatocellular carcinoma	↓ Invasion ↓ Migration	Tumor suppressor	Lv et al., 2017 [65]
hsa-miR-325	MIMAT0000771	Downregulation (tissues)	HMGB1	3′UTR	Non-small cell lung cancer	↓ Invasion ↓ Proliferation	Tumor suppressor	Yao et al., 2015 [66]
hsa-miR-27a-3p	MIMAT0000084	Upregulation (tissues)	Calreticulin	3′UTR	Colorectal cancer	↓ Mitoxantrone and oxaliplain-induced apoptosis ↓ Autophagy ↓ MCH-I expression ↓ Dendritic cell maturation ↓ In situ immune cells infiltration ↑ Tumor growth ↑ Liver metastasis	Onco-miRNA	Colangelo et al., 2016a and b [67,68]
hsa-miR-27b-3p	MIMAT0000419	Not evaluated	Hsp90	3′UTR	Non-small-cell lung carcinoma	↓ Migration ↓ Invasion	Tumor suppressor	Dong et al., 2019 [69]
hsa-miR-34a-3p	MIMAT0004557	Not evaluated	HMGB1	3′UTR	Retinoblastoma	↑ Etoposide and carboplatin-induced apoptosis ↓ Starvation-induced autophagy	Tumor suppressor	Liu et al., 2014 [70]
hsa-miR-34a-5p	MIMAT0000255	Downregulation (cell lines)	HMGB1	3′UTR	Acute Myeloid Leukemia	↑ Apoptosis ↓ Autophagy	Tumor suppressor	Liu et al., 2017 [58]
		Downregulation (tissues)		3′UTR	Cervical (CaCx) and colorectal (CRC) cancers	↓ Invasion ↓ Migration ↓ Proliferation	Tumor suppressor	Chandrasekaran et al., 2016 [71]
		Downregulation (tissues and cell lines)		3′UTR	Cutaneous squamous cell carcinoma	↓ Invasion ↓ Migration ↓ Proliferation	Tumor suppressor	Li et al., 2017 [72]
hsa-miR-361-5p	MIMAT0000703	Downregulation (tissues and cell lines)	Hsp90	3′UTR	Cervical cancer	↓ Epithelial-mesenchymal transition ↓ Invasion	Tumor suppressor	Xu et al., 2020 [73]
hsa-miR-449a	MIMAT0001541	Downregulation (tissues and cell lines)	HMGB1	3′UTR	Non-small cell lung cancer	↓ Invasion ↓ Migration ↓ Proliferation	Tumor suppressor	Wu et al., 2019 [74]
hsa-miR-505-3p	MIMAT0002876	Downregulation (cell lines)	HMGB1	3′UTR	Gastric cancer	↓ Migration ↓ Proliferation ↑ Apoptosis	Tumor suppressor	Tian et al., 2018 [75]
		Downregulation (tissues and cell lines)		3′UTR	Hepatocellular carcinoma	↓ Proliferation ↓ Invasion ↓ Epithelial-mesenchymal transition	Tumor suppressor	Lu et al., 2016 [76]
		Not evaluated		Non evaluated	Hepatocellular carcinoma	↑ Doxorubicin-induced apoptosis	Tumor suppressor	Lu et al., 2018 [77]
		Downregulation (tissues)		3′UTR	Osteosarcoma	↓ Invasion ↓ Migration ↓ Proliferation	Tumor suppressor	Liu et al., 2017 [78]
hsa-miR-519d-3p	MIMAT0002853	Downregulation (tissues)	HMGB1	3′UTR	Lung cancer	↓ Invasion ↓ Migration ↓ Proliferation	Tumor suppressor	Ye et al., 2020 [79]
hsa-miR-548b-3p	MIMAT0003254	Downregulation (tissues and cell lines)	HMGB1	3′UTR	Hepatocellular carcinoma	↓ Invasion ↓ Migration ↓ Proliferation ↑ Apoptosis	Tumor suppressor	Yun et al., 2019 [80]
hsa-miR-628-3p	MIMAT0003297	Not evaluated	Hsp90	3′UTR	Non-small-cell lung carcinoma	↓ Migration ↑ Apoptosis	Tumor suppressor	Pan et al., 2018 [81]
hsa-miR-665	MIMAT0004952	Downregulation (tissues and cell lines)	HMGB1	3′UTR	Retinoblastoma	↓ Invasion ↓ Migration ↓ Proliferation ↑ Apoptosis	Tumor suppressor	Wang et al., 2019 [82]

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
