# Peer review of "Damage-Associated Molecular Patterns Modulation by microRNA: Relevance on Immunogenic Cell Death and Cancer Treatment Outcome"

_cancers, 2021, doi:10.3390/cancers13112566_

Round 1

Reviewer 1 Report

In this review article by Lamberti et al., the authors discuss the potential role of miRNAs in modulating immunogenic cell death (ICD) in anti-tumor immunity. Considering the clinical relevance of ICD and DAMPs in the context of chemotherapies, radiotherapy, oncolytic virus therapy and photodynamic therapy has been well established, it is certainly an important topic to cover. With that being said, while the authors have given a comprehensive review of previous work demonstrating many DAMPs that are regulated by various miRNAs, I feel that every studies that have been discussed in this manuscript suggested the impact of miRNAs on tumor growth has very little to do with ICD despite the fact that DAMPs are being targeted. Therefore, while I do agree with the authors that the contribution of cell‐autonomous features, like miRNAs, to the development of ICD has not been addressed in sufficient depth, without having enough literature supporting or even implying a role of miRNA in promoting or inhibiting ICD, I am not certain whether this review has achieved the goal they set up originally.

Below are some examples:  

  1. The role of the discussed miRNAs on the inhibition of DAMPs did not seem to have anything to do with ICD. After all, if DAMPs do function as ICD-inducers, overexpression of miRNAs that directly target DAMPs like Hsps should function as oncomiRs rather than tumor suppressors. However, most if not all, DAMP-targeting miRNAs discussed in this manuscript are tumor suppressors.
  2. Similarly, as the authors also acknowledged, despite the potential immunostimulatory role of HMGB1 serving as a DAMP, most studies suggested it (along with Hsps) plays a more important role in promoting tumor proliferation, migration, invasion and chemoresistance. Therefore, even though many miRNAs have been shown to target HMGB1, they do not really play a part in ICD.
  3. Here is another example: Even though administration of miR-200c and doxorubicin promotes ICD and anti-tumor immunity, the fact that miR-200c directly inhibits HMGB1 expression, raising the question as to whether the anti-tumor effect of miR-200c is through targeting the DAMP-ICD axis. Supporting this notion, the authors also acknowledged that miR-200c downregualtes PD-L1 expression further suggesting an ICD-independent anti-tumor role of miR-200c.

Author Response

We are appreciative of the reviewer’s encouragement and the opportunity to revise the manuscript. As he/she pointed out, the contribution of miRNAs to immunogenic cell death’s regulation has not been yet addressed in sufficient depth. Given that DAMPs, such as HMGB1, calreticulin and HSPs, are considered the main hallmarks of ICD, we aimed here to provide a comprehensive and exhaustive overview of the effects of miRNAs on their expression, to help identify commonalities and differences highlighting possible interrelationships between miRNA, DAMPs and ICD. As we mentioned in Conclusion, we expected to identify several oncomiRNAs targeting DAMPs whose ectopic inhibition could promote and/or restore the immunogenic potential of therapeutic agents. Surprisingly, our hypothesis was only valid for CRT targeting, whereas miRNA-based downregulation of other DAMPs was in fact associated with suppressed cancer properties. Following the reviewer’s comments, to more properly specify the content and scope of our review, we have decided to change the title and introduction section. We consider that this new title: “Damage-associated molecular pattern modulation by micro-RNA: relevance on immunogenic cell death and cancer treatment outcome” states the implications of our work more clearly.

Reviewer 2 Report

The Review entitled “Immunogenic cell death hallmark modulation by microRNA and its relevance on cancer treatment outcome" is a well-written paper, the introduction is detailed, the strategy of proposing an overview of microRNAs involved in ICD regulation is clear, new, and well performed. 

Author Response

We appreciate the reviewer for his/her kind comments about the manuscript

Reviewer 3 Report

This is a generally well-written article on miRNA and tumor immunity. This topic is of great interest. The author can further make this paper better; please see below – just a few points for improvements.

The authors discuss how to improve treatment response but ignore influences of other factors. There are many environmental and lifestyle factors that influence miRNA expression (in different cells), immune cells, the microbiota, tumor development and response to therapy. The authors should discuss those points. Many factors influence response to therapy in each patient differentially.

Along with these points, research on environment factors, microbiome, immunity, and molecular tissue biomarkers should be pursued. The authors should discuss molecular pathological epidemiology, which can integrate those factors, molecular pathologies, immune response, and clinical outcomes in cancer. Molecular pathological epidemiology is discussed in Gut 2018, Annu Rev Pathol 2019, etc. MPE research can be a promising direction and improve prediction of response to (immuno-)therapy. 

Author Response

We would like to thank the reviewer for his/her thoughtful review of the manuscript. This input is very helpful and raises a very important issue. Following his/her suggestion, we have decided to include some statements in Section 7 of the manuscript regarding how the application of “molecular pathological epidemiology” research frameworks would impact on our knowledge about miRNA/ICD association in cancer, and its clinical outcome.

Reviewer 4 Report

The manuscript entitled:"Immunogenic cell death hallmark modulation by microRNA and its relevance on cancer treatment outcome" focused on a systemic revision of literature data about the analysis of miRNA impact in the iimprovement in cancer cells immunogenicity represents an interesting paper able to explain literature data about this emerging topic. In my opinion minor considerations regarding the elucidation of biological and clinical mechanisms should be implemented to accept this paper for publication.

  • The authors report the main mechanisms used to regulate gene expression. As regards, please, could the authors show if long non coding RNA may also play a role in the ICD by durectly interacting with miRNA elucidated in this review?
  • The authors well explain biological behaviour of a list of miRNA for ICD in severalcancer types. In this scenario, i would reccomend to focus on emerging clinical studies where these miRNA are considered investigated biomarkers
  • Promoter methylation may be also considered as another pivotal mechanism involved in the gene regulation. Please, could the authors elucidate if this mechanism may also play a pivotal role in ICD for tumor cells?

Author Response

  • The authors report the main mechanisms used to regulate gene expression. As regards, please, could the authors show if long non coding RNA may also play a role in the ICD by directly interacting with miRNA elucidated in this review?

Answer: We agree with the reviewer that this information would have a great value. In section 4, we have summarized and discussed the interaction between several long non coding RNA with DAMPs-targeting miRNAs, such as KCNQ1OT1::miR‐27b‐3p (for Hsp90), NEAT1::miR-361-5p (for Hsp90), UCA1::miR-193a-3p (for HMGB1), TP73-AS1::miR-142-3p and TP73-AS1::miR-200a-5p (for HMGB1), among others. However, those reports did not analyze the impact of these processes on ICD. Unfortunately, as far as we know, there is not enough evidence supporting a role of lncRNA in promoting or inhibiting ICD. This represents a great avenue for future research.

  • The authors well explain biological behaviour of a list of miRNA for ICD in several cancer types. In this scenario, i would reccomend to focus on emerging clinical studies where these miRNA are considered investigated biomarkers

Answer: We fully agree with this comment and appreciate the reviewer´s suggestion. For this reason, we included a new column in Table I showing the information regarding the dysregulation of each miRNA on tumor clinical samples and/or cell lines, reported in the literature and here summarized. This new perspective is now also explicitly mentioned in section 4.1 and 4.3.

  • Promoter methylation may be also considered as another pivotal mechanism involved in the gene regulation. Please, could the authors elucidate if this mechanism may also play a pivotal role in ICD for tumor cells?

Answer: We agree with the reviewer that epigenetic modulation is extremely relevant when studying immunogenic cell death. As mentioned in the Introduction section, a detailed review of the evidence linking epigenetic modifications and ICD has been offered by others (Cruickshank et al. 2018, Ref #8). In their review, they discussed about DNA methylation in cancer and its impact on DAMPs and DAMPs receptor regulation. Given this exhaustive previous work, in our manuscript, we decided to focus specifically on the current knowledge of miRNAs targeting DAMPs.

Round 2

Reviewer 3 Report

The authors improved the paper. 

Reviewer 4 Report

The manuscript may be accepted in the present form